# A Multigraph-Defined Distribution Function in a Simulation Model of a Communication Network

**DOI:** 10.3390/e24091294

**Published:** 2022-09-14

**Authors:** Slobodan Miletic, Ivan Pokrajac, Karelia Pena-Pena, Gonzalo R. Arce, Vladimir Mladenovic

**Affiliations:** 1Electronic Systems Department, Military Technical Institute, 11000 Belgrade, Serbia; 2Department of Electrical and Computer Engineering, University of Delaware, Newark, DE 19716, USA; 3Faculty of Technical Sciences Cacak, University of Kragujevac, 34000 Kragujevac, Serbia

**Keywords:** communication network, multigraphs, adjacency matrix, network simulation, network traffic, distribution function

## Abstract

We presented a method based on multigraphs to mathematically define a distribution function in time for the generation of data exchange in a special-purpose communication network. This is needed for the modeling and design of communication networks (CNs) consisting of integrated telecommunications and computer networks (ITCN). Simulation models require a precise definition of network traffic communication. An additional problem for describing the network traffic in simulation models is the mathematical model of data distribution, according to which the generation and exchange of certain types and quantities of data are realized. The application of multigraphs enabled the time and quantity of the data distribution to be displayed as operational procedures for a special-purpose communication unit. A multigraph was formed for each data-exchange time and allowed its associated adjacency matrix to be defined. Using the matrix estimation method allowed the mathematical definition of the distribution function values. The application of the described method for the use of multigraphs enabled a more accurate mathematical description of real traffic in communication networks.

## 1. Introduction

The design of communication networks as a spatially distributed integrated telecommunication and computer network (ITCN) has been improved by the application of computer simulations. Defining a simulation model of an ITCN is realized by using advanced simulation software with integrated tools. These tools allow an analysis of the network elements’ parameters. The application of this methodology of ITCN simulation model design requires a precise definition of network traffic in addition to a definition of the active and passive elements of the architecture and network topology [1]. Network traffic is the process of time events generating a certain type and amount of data at a source, and their distribution between sources and destinations connected in a communication network. There may be multiple data sources in a network that generate the same or different types and amounts of data at the same or different times, and destinations may simultaneously receive data from one or more sources. The problem in designing a simulation model is generating an accurate description of this network traffic. Depending on the purpose of the communication network, the network traffic process can be a deterministic or stochastic event.

In accordance with this definition, network traffic requires the application of appropriate models and distribution functions of communication data over time. An additional problem of describing network traffic in the simulation model is defining the mathematical model according to which the generation of and change in the amount of data are realized. The model requires an appropriate distribution function that, in the simulation model, temporally describes the generation and distribution of the amount of data between network elements. We used the sampling matrix associated with multigraphs [2] to derive the time distribution function of the communication events of network traffic. A method of applying multigraphs for defining the distribution functions of the generation time of data between network elements of the ITCN is presented in this study. The contributions of this study are as follows:A new method of defining network traffic was proposed. The distribution function for creating a simulation model of a communication network was developed, based on the description of communication events and the values of the parameters they determined. The application of this method enabled us to solve the problem of describing the time of data generation and distribution in the communication networks.The application of multigraphs for the mathematical derivation of a more precise distribution function of data was proposed and compared with other methods in which the distribution function of data was approximated by the type of network traffic and by the time variation of the data.The application of multigraphs and their related matrices enabled multiple descriptions of network traffic in terms of events and communication parameters, which enabled their change in time to be mathematically represented as a function of the schedule. The new approach enabled a more accurate description of the network traffic in the design of a simulation model of the communication network and time-accurate results in the simulation.

The paper is organized as follows. Section 2 provides an overview of the different methods used for defining and statistically describing network traffic. Definitions of all the starting elements needed to describe network communications are presented in Section 3. In Section 4, the basic concept, and details of the proposed method of applying multigraphs for describing the time of network traffic and data distribution are presented. Section 5 presents an application of the mathematical derivation, and a graphical representation of the time distribution functions in the proposed method. Section 6 concludes the study and gives directions for further research in the application of the method.

## 2. Related Work

In earlier works, different methods of defining network traffic were proposed. Network traffic is a complex time-stochastic or -deterministic process of network structure. Earlier methods consisted of complex procedures for describing and defining the network traffic. The methods are complex, especially for describing network traffic with the distribution of multiple data formats. The basic method of defining network traffic was realized by measuring and recording traffic in test networks, as in [3,4,5,6,7], with the theoretical derivation of statistical mathematical descriptions for further analysis. Measurements of the generated and distributed network traffic enabled statistical descriptions and parametric descriptions in [4,5,8]. In recent work [3,6,7,9,10,11,12,13], statistical typification of network traffic with known distribution functions and parameter variation was defined or the traffic was described by using self-similarity related to heavy-tail distributions [14,15]. Further definitions of network traffic were limited to the recognition network traffic type (voice internet, HTTP, VoIP, multimedia, etc.) and descriptions of the intended distribution function. An overview and comparison of the methods used in previous studies to define network traffic are given in Table 1.

These methods are integrated into simulation tools such as OPNET and other advanced software simulation packages. The application of the methods described in previous research may lead to incorrect selection or description of the statistical parameters of the distribution function. The consequence is that one may obtain incorrect simulation results and derive erroneous conclusions and decisions about the design of the network structure. To increase the accuracy of parameters when describing and defining network traffic in [2], we performed an analytical method where we used multigraphs to describe communication interactions as events between network elements. This method was executed and tested on the example of deterministic arranged communication in the network.

The network traffic matrix model has a significant role in network design, network traffic design and analysis of the results, as in the method given in [18,19]. In the method proposed in this study, we introduced a new approach to defining the basis required to obtain a mathematical model from the network traffic matrix. By applying the mathematical models given in [20,21], the time dimension of the multigraph was added and the estimated distribution function that describes the network traffic as a statistical time event can be obtained.

## 3. Data Exchange in the Communication Network

Central to definition of network traffic models is the matrix of network traffic between the source and the destination of communications in the network. To achieve functional relations between the participants in communication, the type of necessary communication is defined for which information flows are determined. The realization and establishment of information flows in an ITCN require the application of appropriate network application services, marked as (S1, S2, Sn). The data exchange and information are the basis for defining the moment of time t = (t_0_, t_1_, …, t_N−1_, t_N_, t_m_) when the participants in communication establish their communication interaction, achieve mutual communication, and, at the same time, exchange certain types and amounts of data.

The moments of time in which communication should take place, the duration of the communication, and the types of information provided for the data exchanged in communication are the particular operational procedures of communication. In Figure 1, the operational procedures determine the moments of change in the amount of data to be distributed between elements of the organizational structure. The type of data and the information (voice, message, symbol, text, table, image, video, etc.) to be exchanged between participants in the communication process are defined by the operational procedures for the command operational function. Starting from the defined communication relationships between the elements of the organization, the matrix of network traffic can be obtained.

In accordance with the concepts given in [2], communication data exchanges are defined. The time distribution of the data is mapped to the distribution function for the OPNET simulation model, which is shown by the logical flow in Figure 1.

### 3.1. The Data of Network Distribution over Time

Designing a simulation model required us to define the variation in the amount of data generated at the source sent to the destination. The value of the amount of data (Adt) in kbps or Mbps was determined for each type of information (voice, message, symbol, text, table, database, image, video, etc.) exchanged between the elements.

The distribution of the amount of data occurs at the moments of time t = (t_0_, t_1_, …, t_N−1_, t_N_, t_m_). Additionally, depending on the function in the communication process, the minimum and maximum amount of data generated by the network element, Ei, for distribution in the network is calculated. The transfer of information to the ITCN requires encrypting of the communication channels. The amount of data for distribution in the network can be increased by the amount of digital code required for protecting information (reconstruction, encryption, error detection). The increase in the amount of data is realized in relation to the header size of the individual layers of the OSI network model. The choice of the access technique, the technology and transmission medium, the communication protocols, and the data packet size (MTU) affect the amount of data to be transferred by the telecommunication links in the ITCN. The total data payload for distribution by the network from the source to destination is determined by the steps in the procedure shown in Figure 2.

The distribution function of communication interactions between network elements according to the method given in [2] represents the law of data generation over time. Data generation is realized by the application services.

### 3.2. Distribution Function for Variations in the Amount of Data

The accuracy of the network traffic simulation results in the OPNET simulation model is conditioned by the choice of the distribution function. The distribution function should describe the generation of data and the variation in the amount of data over time. The selection of the distribution function requires one to define its parameters. Moreover, the distribution function is based on the statistical study of communication in the network traffic record, as in [3,4,5,6,7,11,12,13], or is based on an approximation of the type of network traffic (audio, messages, text, IP, VoIP, video, HTTP, web, ATM, etc.) with existing known distribution functions (exponential, Poisson, Normal (Gaussian), uniform, Weibull etc.), as in [8,10,16,17]. Data traffic modeling is based on self-similarity with the Pareto distributions and the α-stable distributions, as in [14,15]. Two basic parameters describe the event of data generation at the source in the simulation model. The data generation time is the first parameter. The time of data generation should be adjusted by the time of establishing communication between the network elements. The variation in the amount of data over the duration of the communication is the second parameter. If one chooses an inappropriate distribution function, or by incorrectly defining the value of the variation in the amount of data, or by incorrectly defining the time, the network traffic will be described incorrectly. Simulations of incorrectly described network traffic will not match the predicted network traffic in an ITCN. In that case, the simulation results are not accurate for analyzing and optimizing the communication network. The error in describing the amount of data generated over time is reduced when the amount of data generated is defined based on the ITCN network traffic matrix and the corresponding distribution function.

## 4. Description of the ITCN Network Distribution Using Multigraphs

The network data distribution in an ITCN is realized based on the operating procedures and according to the methodology specified in [1]. The information flows of the distribution of the predicted types of information between the network elements are described as well. The multigraphs are defined based on this description. The application of multigraphs allows the time relationships of data distribution to be displayed based on the operational communication procedures. For each moment of time t = (t_0_, t_1_, …, t_N−1_, t_N_, t_m_) when data are exchanged, a multigraph is formed. The formed multigraph is joined with the similarity matrix. The corresponding value of the distribution function for each moment of time is calculated by mathematical estimation of the similarity matrix associated with the multigraph. The use of all the calculated values for all moments of time in the communication interval ΔT = [t_0_, t_m_] enables the definition of the appropriate distribution function.

### 4.1. Data Distribution Time Scheme between ITCN Network Elements

The generation and distribution of data between network elements in the ITCN are realized through the network application services Srv, rv = (1, 2, …). Each application service in the ITCN enables the establishment of communication and network distribution of the appropriate type of data. The Srv application service on the network element Ei is activated by establishing a communication interaction between the network elements Ei and Ej (i ≠ j) at the moments of time t = (t_0_, t_1_, …, t_N−1_, t_N_, t_m_). The moments of time are set at the beginning of the time interval in which the application service is active between the network elements. The generation of communication information is enabled and transformed into the appropriate amount of digital data for distribution to the network element Ej is performed. The time scheme of communication interactions (Figure 3), as in [1,2], shows the flow of these activities from the operational procedures.

The individual timeline of the individual service Srv now of time t = (t_0_, t_1_, …, t_N−1_, t_N_, t_m_) of activation are separated from the given time scheme. The amount of data Adt = (Adt_0_, Adt_1_, …, Adt_N−1_, Adt_N_, Adt_m_) generated at the moment of time t = (t_0_, t_1_, …, t_N−1_, t_N_, t_m_) in the application service Srv is also defined. Examples of the separate individual timing schemes for Services S1 and S2 are displayed in Figure 4.

Other ways of representing the communication activation of application services S1 to S4 between network elements E1 to E8 are shown in Figure 5. This representation is used to define the amount of data generated Adt at the moment of time t = (t_0_, t_1_, …, t_N−1_, t_N_, t_m_). For example, the service S1 in the network element E1 with a data quantity of Adt_0_ = 15 kbps for distribution to the network element E2 at time t_0_ is denoted as E1E2S1_Adt_0_.

### 4.2. Multigraphs of Data Distribution in ITCN Network Traffic

The data exchanged by the applicable service Srv at each moment of time t = (t_0_, t_1_, …, t_N−1_, t_N_, t_m_) are shown by presenting the network traffic as a multigraph (Figure 6a). The single-service multigraph (labeled SSMG_Srv_Adt) shows the amount of data Adt (kbps or Mbps) exchanged between the network elements Ei and Ej (i ≠ j) by the application service Srv at time t. A single edge between the nodes (simple graphs) Ei and Ej (i ≠ j) represents the communication interaction between these network elements, where the amount of data Adt are distributed through the application service Srv at time t. The creation of all the single-service multigraphs between the network elements Ei and Ej (i ≠ j) through the application service Srv for each moment of time t = (t_0_, t_1_, …, t_N−1_, t_N_, t_m_) enables the presentation of data exchanged during the communication time interval ΔT = [t_0_, t_m_]. The total data exchanged between the nodes Ei and Ej (i ≠ j) through all application services, are Srv = (S1, S2, …, Sn) with time t representing the unification of all the single-service multigraphs formed previously into one multi-service multigraph (labeled MSMG_S1Sn_Adt), as shown in Figure 6b. The multi-service multigraph enables the definition of network traffic among the ITCN’s network elements at the observed moments of time t = (t_0_, t_1_...t_N−1_, t_N_, t_m_) and enables the application of graph sampling theory to perform predictions, as in [22].

The creation of multi-service multigraphs for each moment of time t = (t_0_, t_1_, …, t_N−1_, t_N_, t_m_) enables the presentation of the exchange of all data through all application services during the communication time interval ΔT = [t_0_, t_m_]. A set of multi-service multigraphs allows one to define the total network traffic among the ITCN’s network elements during the communication time interval ΔT = [t_0_, t_m_].

### 4.3. Matrix Associated with the ITCN Network Traffic Distribution Multigraph

The multigraph data distribution in network traffic of the ITCN is mathematically represented by the symmetric matrix T_SSMG_Srv_Adt_ in Equation (1) with integer terms and a diagonal of zero, where n is the number of network elements Ei. The associated symmetric matrix is formed by using a timeline or a time plane of the communication interactions (Figure 5) or by using a single-service multigraph (Figure 6a), such that
(1)$$T_{\mathrm{SSMG}\_\mathrm{Srv}\_\mathrm{Adt}}=\left[\begin{array}{ccccccc} 0 & Ad_{12}t & Ad_{13}t & Ad_{14}t & Ad_{15}t & . & Ad_{1n}t\\ Ad_{21}t & 0 & Ad_{23}t & Ad_{24}t & Ad_{25}t & . & Ad_{2n}t\\ Ad_{31}t & Ad_{32}t & 0 & Ad_{34}t & Ad_{35}t & . & Ad_{3n}t\\ Ad_{41}t & Ad_{42}t & Ad_{43}t & 0 & Ad_{45}t & . & Ad_{4n}t\\ Ad_{51}t & Ad_{52}t & Ad_{53}t & Ad_{54}t & 0 & . & Ad_{5n}t\\ Ad_{61}t & Ad_{62}t & Ad_{63}t & Ad_{64}t & Ad_{65}t & . & Ad_{6n}t\\ . & . & . & . & . & 0 & .\\ Ad_{n1}t & Ad_{n2}t & Ad_{n3}t & Ad_{n4}t & Ad_{n5}t & . & 0 \end{array}\right]$$
where Ad_ij_t is the amount of data distributed in the communication interactions between the nodes Ei and Ej (i ≠ j) with the application service Srv = (S1, S2, Sn) at the moment of time t = (t_0_, t_1_, …, t_N−1_, t_N_, t_m_). Figure 7 shows the single-service multigraph for data exchanged among the network elements E1 to E8 with the application service S1 at the moment of time t_0_, and its associated symmetric 8 × 8 matrix.

For all single-service multigraphs, the set of associated matrices T_SSMG_Srv_Adt_ defines the matrix at each moments of time t = (t_0_, t_1_, …, t_N−1_, t_N_, t_m_), as in Equations (2) and (3):
(2)$$T_{\mathrm{SSMG}\_\mathrm{S1}\_{\mathrm{Adt}}_{0}}=\left[\begin{array}{ccccccc} 0 & Ad_{12}t_{0} & Ad_{13}t_{0} & Ad_{14}t_{0} & Ad_{15}t_{0} & . & Ad_{1n}t_{0}\\ Ad_{21}t_{0} & 0 & Ad_{23}t_{0} & Ad_{24}t_{0} & Ad_{25}t_{0} & . & Ad_{2n}t_{0}\\ Ad_{31}t_{0} & Ad_{32}t_{0} & 0 & Ad_{34}t_{0} & Ad_{35}t_{0} & . & Ad_{3n}t_{0}\\ Ad_{41}t_{0} & Ad_{42}t_{0} & Ad_{43}t_{0} & 0 & Ad_{45}t_{0} & . & Ad_{4n}t_{0}\\ Ad_{51}t_{0} & Ad_{52}t_{0} & Ad_{53}t_{0} & Ad_{54}t_{0} & 0 & . & Ad_{5n}t_{0}\\ Ad_{61}t_{0} & Ad_{62}t_{0} & Ad_{63}t_{0} & Ad_{64}t_{0} & Ad_{65}t_{0} & . & Ad_{6n}t_{0}\\ . & . & . & . & . & 0 & .\\ Ad_{i1}t_{0} & Ad_{i2}t_{0} & Ad_{i3}t_{0} & Ad_{i4}t_{0} & Ad_{i5}t_{0} & . & 0 \end{array}\right]$$
(3)$$T_{\mathrm{SSMG}\_\mathrm{S1}\_{\mathrm{Adt}}_{m}}=\left[\begin{array}{ccccccc} 0 & Ad_{12}t_{m} & Ad_{13}t_{m} & Ad_{14}t_{m} & Ad_{15}t_{m} & . & Ad_{1n}t_{m}\\ Ad_{21}t_{m} & 0 & Ad_{23}t_{m} & Ad_{24}t_{m} & Ad_{25}t_{m} & . & Ad_{2n}t_{m}\\ Ad_{31}t_{m} & Ad_{32}t_{m} & 0 & Ad_{34}t_{m} & Ad_{35}t_{m} & . & Ad_{3n}t_{m}\\ Ad_{41}t_{m} & Ad_{42}t_{m} & Ad_{43}t_{m} & 0 & Ad_{45}t_{m} & . & Ad_{4m}t_{0}\\ Ad_{51}t_{m} & Ad_{52}t_{m} & Ad_{53}t_{m} & Ad_{54}t_{m} & 0 & . & Ad_{5n}t_{m}\\ Ad_{61}t_{m} & Ad_{62}t_{m} & Ad_{63}t_{m} & Ad_{64}t_{m} & Ad_{65}t_{m} & . & Ad_{6n}t_{m}\\ . & . & . & . & . & 0 & .\\ Ad_{i1}t & Ad_{i2}t & Ad_{i3}t & Ad_{i4}t & Ad_{i5}t & . & 0 \end{array}\right]$$
The set of associated matrices T_SSMG_Srv_Adt_ enables one to define the function for the distribution of data in the network through the service Srv = (S1, S2, …, Sn) in the communication time interval ΔT = [t_0_, t_m_].

The variation in the value of the amount of data VarAdt distributed by the application service Srv = (S1, S2, …, Sn) during the communication time interval ΔT = [t_0_, t_m_] is defined by the minimum and maximum values of the amount of data distributed among the network elements of the ITCN:(4)$$\;VarAdt=\left[Ad_{min},\;Ad_{max}\right]\;$$
(5)$$Ad_{min}=min\left\{Ad_{ij}t_{0},\ldots ,Ad_{ij}t_{m}\right\},\;i=\left[1,n\right]\;\;j=\left[1,n\right]\;i\neq j$$
(6)$$Ad_{max}=max\left\{Ad_{ij}t_{0},\ldots ,Ad_{ij}t_{m}\right\},\;i=\left[1,n\right]\;\;j=\left[1,n\right]\;i\neq j.$$
For the multi-service multigraph, the associated symmetric n × n matrix T_MSMG_S1Sn_Adt_ of data distribution shown in Equation (7) is formed. The value of the distribution function of the total amount of data distributed through all application services Srv at the moments of time t = (t_0_, t_1_, …, t_N−1_, t_N_, t_m_) is defined by the associated symmetric matrix T_MSMG_S1Sn_Adt_.
(7)$$T_{\mathrm{MSMG}\_\mathrm{S1Sn}\_\mathrm{Adt}}=\left[\begin{array}{cccccc} 0 & sAd_{12}t & sAd_{13}t & sAd_{14}t & . & sAd_{1n}t\\ sAd_{21}t & 0 & sAd_{23}t & sAd_{24}t & . & sAd_{2n}t\\ sAd_{31}t & sAd_{32}t & 0 & sAd_{34}t & . & sAd_{3n}t\\ sAd_{41}t & sAd_{42}t & sAd_{43}t & 0 & . & sAd_{4n}t\\ sAd_{51}t & sAd_{52}t & sAd_{53}t & sAd_{54}t & . & sAd_{5n}t\\ sAd_{61}t & sAd_{62}t & sAd_{63}t & sAd_{64}t & . & sAd_{6n}t\\ . & . & . & . & 0 & .\\ ssAd_{n1}t & sAd_{n2}t & sAd_{n3}t & sAd_{n4}t & . & 0 \end{array}\right]$$
where sAd_ij_t is the total amount of data distributed in communication interactions between nodes Ei and Ej (i ≠ j) with all the activated application services Srv = (S1, S2, …, Sn) at the moments of time t = (t_0_, t_1_, …, t_N−1_, t_N_, t_m_), where:(8)$$sAd_{ij}t=\overset{Sn}{\underset{S1}{\sum }}\overset{n}{\underset{i=1}{\sum }}\overset{n}{\underset{j=1}{\sum }}Ad_{ij}t\;,\;\;i\neq j,\;t=\left(t_{0,\;\;}\;t_{1\;\ldots \;}t_{N-1,\;\;}t_{N,\;\;}t_{m}\right)\;\;$$
The set of associated matrices T_MSMG_S1Sn_Adt_ at each moment of time t = (t_0_, t_1_, …, t_N−1_, t_N_, t_m_) defines a set of multi-service multigraphs. The set of associated symmetric matrices T_MSMG_S1Sn_Adt_ enables the definition of the value of the distribution function of the total amount of data distributed through all the application services Srv during the communication time interval ΔT = [t_0_, t_m_], such that
(9)$$T_{\mathrm{MSMG}\_\mathrm{S1Sn}\_{\mathrm{Adt}}_{0}}=\left[\begin{array}{cccccc} 0 & sAd_{12}t_{0} & sAd_{13}t_{0} & sAd_{14}t_{0} & . & sAd_{1n}t_{0}\\ sAd_{21}t_{0} & 0 & sAd_{23}t_{0} & sAd_{24}t_{0} & . & sAd_{2n}t_{0}\\ sAd_{31}t_{0} & sAd_{32}t_{0} & 0 & sAd_{34}t_{0} & . & sAd_{3n}t_{0}\\ sAd_{41}t_{0} & sAd_{42}t_{0} & sAd_{43}t_{0} & 0 & . & sAd_{4n}t_{0}\\ sAd_{51}t_{0} & sAd_{52}t_{0} & sAd_{53}t_{0} & sAd_{54}t_{0} & . & sAd_{5n}t_{0}\\ sAd_{61}t_{0} & sAd_{62}t_{0} & sAd_{63}t_{0} & sAd_{64}t_{0} & . & sAd_{6n}t_{0}\\ . & . & . & . & 0 & .\\ sAd_{n1}t_{0} & sAd_{n2}t_{0} & sAd_{n3}t_{0} & sAd_{n4}t_{0} & . & 0 \end{array}\right]$$
(10)$$T_{\mathrm{MSMG}\_\mathrm{S1Sn}\_{\mathrm{Adt}}_{m}}=\left[\begin{array}{cccccc} 0 & sAd_{12}t_{m} & sAd_{13}t_{m} & sAd_{14}t_{m} & . & sAd_{1n}t_{m}\\ sAd_{21}t_{m} & 0 & sAd_{23}t_{m} & sAd_{24}t_{m} & . & sAd_{2n}t_{m}\\ sAd_{31}t_{m} & sAd_{32}t_{m} & 0 & sAd_{34}t_{m} & . & sAd_{3n}t_{m}\\ sAd_{41}t_{m} & sAd_{42}t_{m} & sAd_{43}t_{m} & 0 & . & sAd_{4n}t_{m}\\ sAd_{51}t_{m} & sAd_{52}t_{m} & sAd_{53}t_{m} & sAd_{54}t_{m} & . & sAd_{5n}t_{m}\\ sAd_{61}t_{m} & sAd_{62}t_{m} & sAd_{63}t_{m} & sAd_{64}t_{m} & . & sAd_{6n}t_{m}\\ . & . & . & . & 0 & .\\ sAd_{n1}t_{m} & sAd_{n2}t_{m} & sAd_{n3}t_{m} & sAd_{n4}t_{m} & . & 0 \end{array}\right]$$
The elements for defining the data distribution function are realized by forming all the sets of associated single-service matrices T_SSMG_Srv_Adt_ and all the sets of the associated multi-service matrices T_MSMG_S1Sn_Adt_.

## 5. Generating the Data Distribution Function in the ITCN by Sampling Multigraphs

The function distribution of the amount of data in the time interval (pdF(ΔT)) for implementation in the OPNET simulation model is defined by sampling the single-service and multi-service multigraphs of data distribution. Sampling the multigraphs is equivalent to sampling the associated symmetric matrices given in [23]. The associated symmetric matrix is sampled using the sequential importance sampling (SIS) method for sampling multigraphs given in [20]. The value of the estimated distribution function represents the content of the multigraph and is determined by applying the asymptotic approximation given in [20,21].

Additionally, the distribution function for the approximation of multigraphs is defined by using graphs and weighting coefficients and applying the methods given in [20,24].

The matrix T_SSMG_(t) = T_SSMG_Srv_Adt_ belongs to the set of associated symmetric matrices related to the distribution of data between the network elements Ei and Ej (i ≠ j) with the application service Srv at the moments of time t = (t_0_, t_1_, …, t_N−1_, t_N_, t_m_), where ΣT is the number of matrices in the set. The distribution function q(T_SSMG_(t)) > 0 for the matrix T_SSMG_(t) defines the amount of data for distribution between the network elements Ei and Ej (i ≠ j) through the application service Srv at the moments of time t = (t_0_, t_1_, …, t_N−1_, t_N_, t_m_). The estimated value of the distribution function is:(11)$$E_{q}\left[\frac{1}{q\left(T_{SSMG}\left(t\right)\right)}\right]=\underset{T}{\sum }\;\frac{1}{q\left(T_{SSMG}\left(t\right)\right)}\;q\left(T_{SSMG}\left(t\right)\right)=\left|\sum^{\;}T\right|$$
(12)$$\left|\sum^{\;}T\right|=\frac{1}{n}\sum_{i=1}^{n}\frac{1}{q\left(T_{SSMG}\left(t\right)\right)}{\;}_{.}$$
The distribution function q(T_SSMG_(t)) is determined with the test distribution function q(·) by sampling the T_SSMG_(t) matrix column by column (c_1_, c_2_, …, c_n_), using the method and procedure in [20] and [21]. Here, q(T_SSMG_(t)) is represented by:(13)$$q\left(T_{SSMG\;}\left(t\right)=\left(c_{1},c_{2},\ldots c_{n}\right)\right)=\;q\left(c_{1}\right)q\left(c_{1}|c_{2}\right)\ldots q\left(c_{n}|c_{n-1}\ldots ,c_{1}\right)t_{1},\;\ldots ,\;t_{N-1}$$
The sum of the row margin (d_i_) of the matrix (n × n), denoted d^(2)^, d^(3)^, …, d^(i)^, and the updated row margins of the (n−1) × (n−1) submatrix are determined for the matrix T_Ei_(t).

The procedure of sampling and removing the matrix columns in T_SSMG_(t) is repeated until all the columns (c_1_, c_2_, …, c_n_) have been sampled. The value of each margin of the row (d_i_) and the total margin of the matrix (M) is calculated according to the following:(14)$$d_{i}=\overset{n}{\underset{j=1}{\sum }}\alpha_{ij}\;,\;\;i=\left[1,n\right]$$
(15)$$d=\left(d_{1},\;d_{2},\;d_{3}\ldots ,d_{n}\right)$$
(16)$$d^{\left(2\right)}=\left(d_{2}-\alpha_{21},\;d_{4}-\alpha_{42},\ldots ,d_{n}-\alpha_{n2}\right)$$
(17)$$d^{\left(i\right)}=\left(d_{i}-\alpha_{i,i-1},\;d_{i+1}-\alpha_{i+1,i-1},\ldots ,d_{n}-\alpha_{n,i-1}\right),\;i=\left[2,n\right]$$
(18)$$\;\;\;M=\overset{n}{\underset{i=1}{\sum }}d_{i}$$
For the T_SSMG_(t) matrix (t), the number of multigraphs |Σd| is calculated. Submatrices are formed by removing columns. For forming the submatrices, the number of multigraphs |Σd^(i)^| is calculated, which corresponds to the associated submatrix. Based on the asymptotic approximation given in [20] and [21], the expression for |Σd| and for |Σd^(i)^| is performed.
(19)$$\left|\sum^{\;}d\right|∼\Delta_{d}\equiv \frac{f\left(M\right)}{\prod_{i=1}^{n}d_{i}!}e^{a\left(d\right)}$$
(20)$$f\left(M\right)=M!/{\left[\left(\frac{M}{2}\right)!2^{\frac{M}{2}}\right]}_{\;}$$
(21)$$a\left(d\right)={\left(\underset{i}{\sum }\left(\begin{array}{c} d_{i}\\ 2 \end{array}\right)/M\right)}^{2}-\underset{i}{\sum }\left(\begin{array}{c} d_{i}\\ 2 \end{array}\right)/M_{\;}$$
(22)$$\left|\sum^{\;}d^{\left(2\right)}\right|∼\Delta_{d^{\left(2\right)}}\equiv \frac{f\left(M-2d_{1}\right)}{\prod_{i=1}^{n}(d_{i}-\alpha_{i1})!}e^{a\left(d^{\left(2\right)}\right)}$$
The expression obtained for each column (c_1_, c_2_, …, c_n_) of the T_SSMG_(t) matrix determines the marginal distribution function of each column p(c_i_) ∼ q(c_i_). The marginal distribution function represents the derived distribution function q(T_SSMG_(t)).
(23)$$p\left(c_{1}=\left(0,\alpha_{21},\ldots ,\;\alpha_{n1}\right)\right)=\frac{\left|\sum^{\;}d^{\left(2\right)}\right|}{\left|\sum^{\;}d\right|}$$
(24)$$p\left(c_{2}\right)=\frac{\left|\sum^{\;}d^{\left(3\right)}\right|}{\left|\sum^{\;}d\right|}$$
(25)$$p\left(c_{n-1}\right)=\frac{\left|\sum^{\;}d^{\left(n\right)}\right|}{\left|\sum^{\;}d\right|}$$
Combining the Expressions in (19) and (22) derives an expression for q(*c*_1_):(26)$$q\left(c_{1}=\left(0,\alpha_{21},\ldots ,\;\alpha_{n1}\right)\right)=\frac{1}{\prod_{i=1}^{n}(d_{i}-\alpha_{i1})!}e^{a\left(d^{\left(2\right)}\right)}$$
The expressions for q(c_1_|c_2_), …, q(c_n_|c_n−1_, …, c_1_) are derived in the same way. The value of q(T_SSMG_(t)) is calculated from the obtained values. The procedure given in [19] evaluates the sampling efficiency of the matrix and the accuracy of the derived distribution function q(T_SSMG_(t)) in relation to the marginal distribution p(T_SSMG_(t)). The value of the standard estimation error μ and the difference between the obtained values of cv^2^ is used to calculate the following expression:
(27)$$\;\hat{\mu }\;=\frac{\sum_{i=1}^{n}f\left(T_{\mathrm{SSMGi}}\left(t\right)\right)\frac{p\left(T_{\mathrm{SSMG}}\left(t\right)\right)}{q\left(T_{\mathrm{SSMG}}\left(t\right)\right)}}{\sum_{i=1}^{n}\frac{p\left(T_{\mathrm{SSMG}}\left(t\right)\right)}{q\left(T_{\mathrm{SSMG}}\left(t\right)\right)}}=\frac{\sum_{i=1}^{n}f\left(T_{\mathrm{SSMG}}\left(t\right)\right)\frac{\frac{1}{\left|q\left(T_{\mathrm{SSMG}}\left(t\right)\right)\right|}}{q\left(T_{\mathrm{SSMG}}\left(t\right)\right)}}{\sum_{i=1}^{N}\frac{\frac{1}{\left|q\left(T_{\mathrm{SSMG}}\left(t\right)\right)\right|}}{q\left(T_{\mathrm{SSMG}}\left(t\right)\right)}}=\;=\frac{\sum_{i=1}^{N}f\left(T_{\mathrm{SSMG}}\left(t\right)\right)\frac{1}{q\left(T_{\mathrm{SSMG}}\left(t\right)\right)}}{\sum_{i=1}^{N}\frac{1}{q\left(T_{\mathrm{SSMG}}\left(t\right)\right)}}$$
The use of weight coefficients (weights) ω_i_ calculated by the procedure given in [20] and [22] realizes the correction and adjustment of values between the derived distribution function q(T_SSMG_(t)) and the marginal distribution p(T_SSMG_(t)).

The application of the previous procedure to all single-service multigraphs and their associated matrices defines the change in the amount of data for distribution in the network. The change in the amount of distributed data is realized between network elements Ei and Ej (i ≠ j) through the application service Srv in the communication time interval ΔT = [t_0_, t_m_]. The calculated values of the distribution function form a set of values of the distribution function q(T_SSMG_(t)). These values enable one to define the data distribution function pdF(Srv(t)) of the application service Srv in the communication time interval ΔT = [t_0_, tm]:(28)$$\left\{q\left(T_{SSMG}\left(t\right)\right)\right\}=>pdF\left(q\left(T_{SSMG}\left(t\right)\right)\right),\;t=\left(t_{0},\;t_{1},\ldots t_{N-1},t_{N},t_{m}\right)$$
(29)$$pdF\left(Srv\left(\Delta T\right)\right)=pdF\left(q\left(T_{SSMG}\left(t\right)\right)\right),\;Srv=\left(S1,\;\ldots Sn\right)\;.$$

The same procedure applies to multi-service multigraphs. The calculated values of the distribution function pdF(q(T_MSMG_(t))) define the data distribution function pdF(S1Sn(ΔT)) of all application services Srv in the communication time interval ΔT = [t_0_, t_m_]. The calculated values form a set of values {q(T_SSMG_(t))}. The use of values from the set of values {q(T_SSMG_(t))} thus formed enables the creation of graphs of the distribution function pdF(Srv(ΔT)). Graphically, the values are connected in the order of the moments of time t = (t_0_, t_1_, …, t_N−1_, t_N_, t_m_) to which the values refer.

The graph of the data distribution function pdF (Figure 8) shows the regularity of the time of the change in the amount of data for distribution among the ITCN’s network elements through the application service Srv in the communication time interval ΔT = [t_0_, t_m_].

The same procedure is used to define the graph of the data distribution function of all the application services pdF(S1Sn(ΔT)) in the communication time interval ΔT = [t_0_, t_m_]. Determining the similarity of the graphs of the function pdF(Srv(t)) to the graphs of the known distribution functions (exponential, Poisson, Normal (Gaussian), uniform, Weibull, etc.) given in [8,10,16,17] allows one to identify the derived distribution function pdF(Srv(t)). The identification of pdF(Srv(t)) as a known distribution function enables the selection of the existing distribution function in the OPNET simulation model and the application of the parameter values from the graphs (Figure 8). If no similarity is found, the use of software tools integrated into the OPNET software allows one to import graphics of the derived distribution function pdF(Srv(t)). This way, the distribution function can be used as a newly defined distribution for realization of the ITCN simulation model.

## 6. Conclusions and Further Research

The application of multigraphs for describing data distribution in an ITCN was described in this study. It enabled a more accurate definition of communication events in the network and a mathematical description of the network traffic. The described method of applying multigraphs is primarily intended for the development of a simulation model of networks with deterministically defined and controlled communication. The method of applying multigraphs is also possible in networks with stochastically generated network traffic, which requires a definition of the variations in the amount of data to be generated between the source and destination. Achieving more accurate results of simulating the predicted communication in an ITCN enables the integration of all the derived distribution functions for a description of the network traffic into the OPNET simulation model. Based on the analysis of the results of realized discrete OPNET simulations of an ITCN, we propose the application of single-service multigraphs, and the derivation and use of the distribution functions of pdF(Srv(t)) of data distribution for each application service. For realizing the simulation of network traffic flows, we propose the use of multi-service multigraphs, and the creation and use of data distribution functions pdF(S1Sn(ΔT)) of all services at the same time. In future research, we will analyze the correlation of the data distribution functions with functions that describe individual network parameters (connectivity, capacity, etc.). This research will enable predictions in the design and optimization of an ITCN using a simulation model.

## Figures and Tables

**Figure 1 entropy-24-01294-f001:**
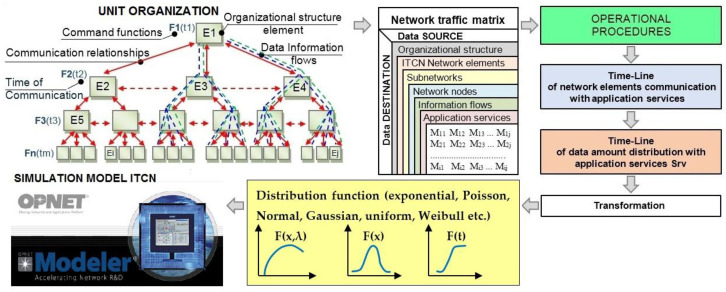
The basic concept of the mapping timeline of the network elements’ distribution of data in the OPNET simulation model.

**Figure 2 entropy-24-01294-f002:**
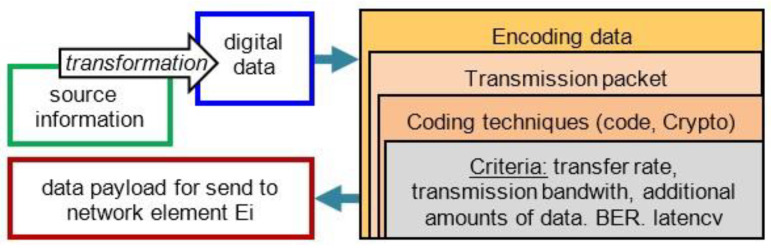
The procedure for determining the amount of data to send to the network elements in the ITCN.

**Figure 3 entropy-24-01294-f003:**
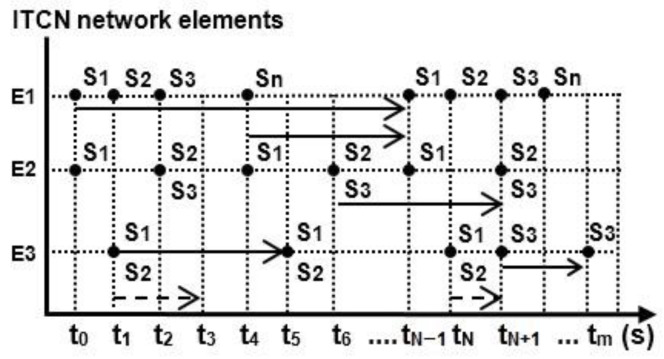
Timeline of activation and repetition of the network elements’ communication interactions and the network application services (Srv).

**Figure 4 entropy-24-01294-f004:**
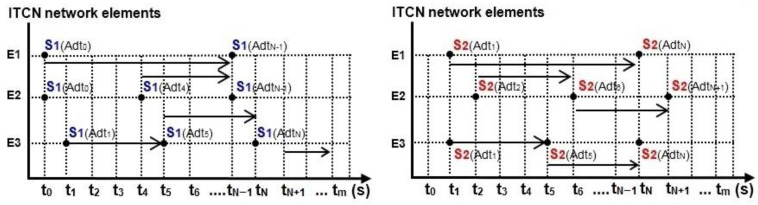
Timeline of activation and repetition of network application services S1 and S2 with amounts of generated data Adt.

**Figure 5 entropy-24-01294-f005:**
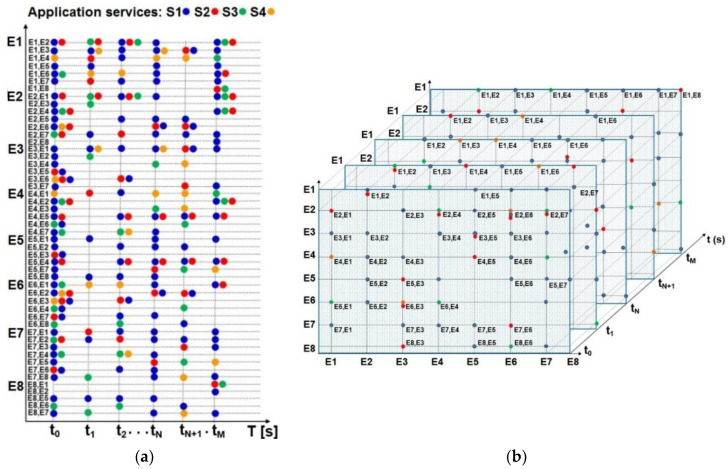
Activation and repetition of the application services S1 to S4 and the communication interactions among network elements E1 to E8: (**a**) timeline; (**b**) time plane.

**Figure 6 entropy-24-01294-f006:**
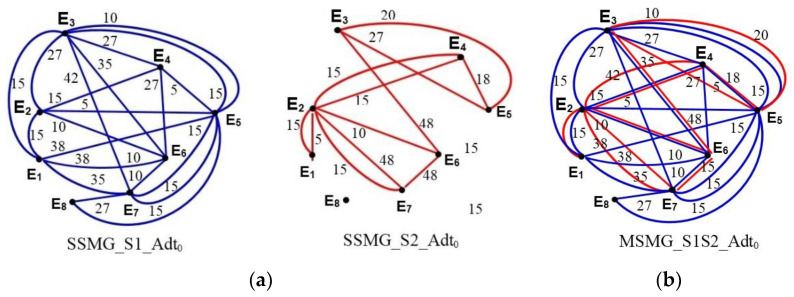
Data exchange multigraph among network elements E1 to E8 with the application services S1 (blue line) to S2 (red line) at time t_0_: (**a**) single-service multigraph; (**b**) multi-service multigraph.

**Figure 7 entropy-24-01294-f007:**
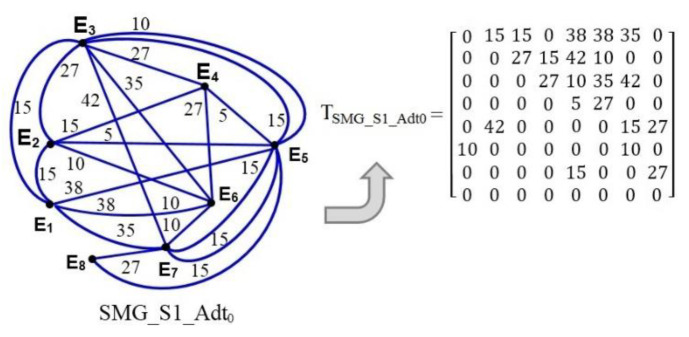
Data exchange single-service multigraph and its symmetric 8 × 8 matrix.

**Figure 8 entropy-24-01294-f008:**
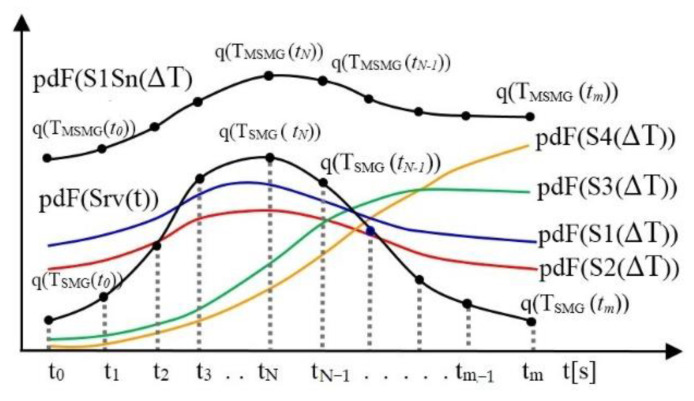
The graph of the data distribution function pdF through the application service Srv in the communication time interval ΔT = [t_0_, t_m_].

**Table 1 entropy-24-01294-t001:** Review of the network traffic defining methods.

Reference	Methods	Measurement Source	Statistical Description	Traffic	Illustrating	Application	Country	Year
[3]	Traffic self-similarity, the approximation function of traffic	The average daily traffic recorded	Pareto distribution	2G, voice, HSDPA,	Function distribution graph	Simulating real network traffic	Russia	2021
[4]	Nonlinear analysis of traffic measurements	A medium-sized LAN with 200 to 250 interconnected computers	Kolmogorov’s scheme for describing network traffic, log-normal distribution, Gaussian distribution	NetBEUI, TCP/IP	Function distribution graph	Realistic dynamical models of network traffic	Russia	2004
[5]	Mathematical approximation	Traffic volume recorded by routers, ethernet traffic traces	Poisson’s probability distribution	Ethernet, MPEG4, TCP/IP, web, email, multimedia	Function distribution graph	Traffic modeling	USA, Texas	2007
[6]	Self-similarity statistical analysis of network traffic measurements	Computer network in small company	Gaussian or power-law probability distributions	Web, HTTP, internet, email, SSL, IPv6	Function distribution graph	Computer network traffic analysis	Poland	2021
[7]	Statistical analysis	Academic, commercial and residential networks; data centers	Log-normal distribution, Gaussian distribution, Weibull distribution	Internet IPv4	Function distribution graph	Predicting the proportion of time traffic, statistically predicted outcomes for the network	USA, Chicago	2019
[8]	Introductory techniques for input modeling; graphical and statistical methods; mathematics	Sample statistics, the Kolmogorov–Smirnov test statistic, the discrete-event simulation, hypothetical arrival process, stochastic processes	Binomial, degenerate Normal, exponential, Bezier curve, independent binomial, bivariate exponential, Markov chain, Poisson process, nonhomogeneous Poisson process, Markov process	Discrete, continuous modeling arrivals	Histogram, function distribution graph	Input models available to simulation analysts	USA	2001
[9]	Simulation modeling process	Describing the behaviors and interactions	Classical statistics right-triangular distribution, cumulative distribution function, uniform distribution	Discrete event systems		Simulating and modeling operations, distribution modeling	USA	2007
[16]	Traffic modeling	Core router of a university, backbone links trans-Pacific backbone link	Gaussian distribution	Gaussian traffic model	Q–Q plots, timescales	Network modeling	Netherlands, Denmark	2013
[10]	Traffic analysis	Counting process, inter-arrival time process, discrete-time traffic,	Poisson, Pareto, Weibull, Markov, Markov chain, on–off model, interrupted Poisson	The traffic on the network	Mathematically, graphs	Traffic modeling, capacity planning the design of networks and services		
[17]	Traffic analysis	The University of Jordan’s network	Poisson traffic model, long-tail traffic models	Internet traffic	Daily traffic flow graph	Traffic model QoS	Jordan	2019
[11]	Traffic analysis, mathematics	1998 FIFA World Cup website	Poisson traffic model, Gaussian distribution	Internet traffic	Function distribution graph	Simulation model	Russia	2017
[12]	Traffic analysis	Hubs of cities in Europe and America	Normal probabilistic	Internet traffic, web traffic	Traffic flow graph, probabilistic distribution	Simulation model	Ukraine	2019
[13]	Traffic analysis	Computer network traffic		Multimedia, VoIP	Average daily computer network traffic	Network modeling	Ukraine	2019

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
