# Peer review of "A Multigraph-Defined Distribution Function in a Simulation Model of a Communication Network"

_entropy, 2022, doi:10.3390/e24091294_

Round 1

Reviewer 1 Report

The paper deals with multigraphs application to develop method for integrated telecommunications and computer networks (ITCN). They describe all the challenges in the design of communication networks, as well as give an overview and explanation of how new concepts are used in the design approach to the use of multigraphs. 

Comment 1:

In line 45-47: There is not entirely clear whether probability of distribution or distribution function is used in the text. Authors should be more precise in terminology. 

Comment 2:

In line 52-55: The sentence "The application of the mathematical method of sampling the matrix associated with the multigraph, which performs the function of the probability of the distribution of moments of time events of network traffic is defined by the application of multigraphs and friend matrices [2]." is too long and not completely clear. It should be corrected and written to be understandable.

Comment 3:

In line 123-125: The terminology in the text is not entirely clear. For example, in the sentence: “The data exchange and information are the basis for defining the time schedule by which the participants in communication establish communication interaction, achieve mutual communication, and at the same time exchange certain types and amounts of data.”, it is not clear what the “time schedule” refers to. Terminology should be uniform throughout the paper.

Comment 4:

In line 129-132: It is not entirely clear what the term “command” refers to? Authors should provide a closer explanation. 

Comment 5:

In line 143-146: The first paragraph in subsection 3.2 is more about the data exchange so, it would be better to move to session 3.

Comment 6:

In lines 196-201 and 246-246: The authors should explain to which time intervals the stated time moments refer

Author Response

The authors are gratefull for the comments of the reviewers that enormously helped quality improvement of the work and this notification will be added in the acknowledgement. As requested, comments of reviewers are provided in received order and carefully answered to the best of authors knowledge with clear notification (remarked as yellow) what has been included in the edited manuscript:

Reviewer I Comments:

The following issues need to be looked at for further improvement of the paper.

Article

Additional editing correction by the authors:

In line 2: The title of the paper editing correction “Multigraphs” to “A multigraph”

Abstract

Additional correction by the authors:

In lines 11-16: To clarify the meanings, we reformulated the sentences to” We present a method based on multigraphs to mathematically define a distribution function in time for the generation of data exchange in a special purpose communication network. This is needed in the modeling and design of a communication network (CN) consisting of integrated telecommunications and computer networks (ITCN). Simulation models require precise definition of the network traffic communication. “

In line 16: Editing correction “the description of” to “describing the”.

In lines 19-20: To clarify the meanings, we reformulated the sentences to” The application of multigraphs enables the time and quantity of the data distribution to be displayed on the basis of operational procedures for the special- purpose communication unit.”

In lines 20-23: We corrected in the sentence “The multigraph” to “A multigraph” and “to be joined” to “to be defined”.

(1) Introduction

Answer: For the introduction improvement we conducted following changes:

Additional correction by the authors:

In lines 34-36: To clarify the meanings, we reformulated the sentences to” The application of the methodology of the ITCN simulation model design requires a precise definition of the network traffic in addition to a definition of the active and passive elements of the architecture and network topology [1].”

Correction according Reviewer I Comment 1.

In lines 45-47: Instead of „models of mathematical functions of the probability of distribution “, we have precisely defined with the „distribution functions of communication data “, which is further used in the rest of the text of the paper.

Additional correction by the authors:

In lines 49-50: To clarify the meaning, we removed from the sentence “the definition of”.

Correction according Reviewer I Comment 2.

In lines 52-57: The long sentence we reformulated with “We used the sampling matrix associated with multigraphs [2] to derive the time distribution function of the communication events of network traffic. A method of applying multigraphs for defining the distribution functions of the generation time of data between network elements of the ITCN is presented in this study. “.

Additional correction by the authors:

In line 59: We replaced the term “paper” with “study”.

In lines 57-58: To clarify the meaning in subsection, we removed the sentence “It is indicated the recognition of the basis for defining the data distribution in the network matrix implemented in the communication network”.

In lines 60-65: To clarify the meaning of the contribution we divide the first sentence on two, and we reformulated with “A new method of defining network traffic is proposed. The distribution function for creating a simulation model of a communication network is developed, based on the description of communication events and the values of the parameters they determine. The application of the method enables us to solve the problem of describing the time of data generation and distribution in the communication networks”. From the sentence in lines 64-65 we removed “and networks of similar purpose”.

In lines 66-69: To simplify the meaning of the contribution in the sentence we corrected “in comparison” to “and compared, and we removed “measurements of” and “in similar networks”. The long sentence we reformulated with “The application of multigraphs for the mathematical derivation of a more precise distribution function of data is proposed and compared with other methods in which the distribution function of data is approximated by the type of network traffic and by the time variation of the data.”.

In lines 70-75: We reformulated the sentence with “The application of multigraphs and their related matrices enables multiple descriptions of network traffic in terms of events and communication parameters, which enables their change in time to be mathematically represented as a function of the schedule. The new approach enables a more accurate description of the network traffic in the design of a simulation model of the communication network and time-accurate results in the simulation.”

(2) Related work

Answer: To clarify the meanings, we redefined the sentences and corrected the terms with more precise ones.

Additional correction by the authors:

In lines 98-99: We correctef in the sentence “papers” to “studies” and “traffic is given” to “traffic are given”.

In lines 113-116: To clarify the meaning we corrected the sentence to “By applying the mathematical models given in [19],[20], the time dimension of the multigraph was added and the estimated distribution function that describes the network traffic as a statistical time event can be obtained.”

(3) Data exchange in the communication network

Answer: To clearly describe the meanings, we redefined the sentences and replaced the terms with more precise ones.

Additional correction by the authors:

In lines 118-119: To clarify the meaning we reformulated the sentence to “Central  to definition of network traffic models is the matrix of network traffic between the source and the destination of communications in the network.”

In lines 121-122: We corrected in the sentence “Realization” to “The realization” and “destination” to “the destination”.

Correction according Reviewer I Comment 3.

In lines 123-125: In order to precisely describe the observed time as a terminology, we replaced the term “time schedule” with the term “the moment of time t = (t0, t1, ...tN-1, tN, tm)”. The defined term has been replaced and is used in the rest of the text of the paper. To clarify the meaning we redefined the sentence to “The data exchange and information are the basis for defining the moment of time t = (t0, t1, ...tN-1, tN, tm) when the participants in communication establish their communication interaction, achieve mutual communication, and at the same time, exchange certain types and amounts of data.”.

Correction according Reviewer I Comment 4.

In lines 129-132: The sentence is supplemented with terms in accordance with Figure 1. In the sentence, we replaced the term “command” with ”communications”, the term ”operating procedures” has been corrected with “operational procedures” and "command function" is clarified by ”the command operational function”.

Additional correction by the authors:

In lines 132-133: We corrected in the sentence “relations” to “relationships” and “ traffic is obtaind” to “ traffic can be obtained”.

In line 134: We corrected in the sentence “concept” to “concepts” and “exchange is defined” to “exchanges are defined”.

In line 140: Editing correction “of mapping” to “of the mapping” and “of network” to “of the network”.

(3.1) Amount of data for network distribution in time (replaced with “The data of network distribution over time”)

Correction according Reviewer I Comment 5.

In lines 143-146:  To make the sentence in paragraph in subsection 3.2 consistent with the data exchange, we have changed the title of the subsection to „The data of network distribution over time“.

Additional correction by the authors:

In lines 144-146:In the sentence we corrected “for exchange between elements” to “exchanged between the elements ”.

In line 147-151: To clarify the meaning we redefined the sentences to “The distribution of the amount of data occurs at the moments of time t = (t0, t1, ...tN-1, tN, tm).”.

In line 157: To clarify the context meaning in the sentence we removed” to be transmitted” and we redefined the sentence with “The total data payload for distribution by the network from source to destination is determined by the steps in the procedure shown in Figure 2.”.

Correction according Reviewer I Comment

In the context of the given comment, we have moved the sentence in line 161-162 to the section move to session 3.

(3.2) Distribution function for data amount variations (Replaced with “Distribution function for variations in the amount of data”)

Additional correction by the authors:

In line 167: We have corrected the title of the subsection with “Distribution function for variations in the amount of data”.

In line 178: We added the new reference “[23]”

(4.) Description of the ITCN network distribution using multigraphs

Additional editing correction:

In line 190: Editing correction in the title of subsection “of ITCN” to “of the ITCN”.

In line 195: We corrected in the sentence “relations” to “relationships” and “of operational” to “of the operational”.

In line 197: We corrected in the sentence “data is exchanged” to “data are exchanged,”

In line 198: To clarify the meaning we redefined the second sentence to “The corresponding value of the distribution function for each moment of time is calculated by mathematical estimation of the similarity matrix associated with the multigraph.”

(4.1) Data distribution time scheme between ITCN network elements

Additional editing correction:

In lines 203-204: We corrected in the sentenece “Generation” to “The generation” and “ITCN” to “the ITCN” and “network” to “the network”.

Correction according Reviewer I Comment 6.

In line 208: In accordance with the given comment, we defined the moments of time with the following sentence „The moments of time are set at the beginning of the time interval in which the application service is active between the network elements.“

Additional editing correction:

In line 209: We corrected in the sentenece “the transformation” to “ transformation”.

In line 211: We corrected in the sentenece “operational ” to “the operational”.

In line 213: We corrected in the title of Figure 3. “Time-Line” to “Time-line” and “network” to “the network”.

In lines 215-216: We corrected in the sentence “time schemes” to “Time-line” and “time of activation t = (t0, t1...tN-1, tN,tm)” to “ moments of time t = (t0, t1...tN-1, tN,tm) of activation”.

In lines 216-217: We removed “which is”.

In lines 218-219: We corrected in the sentence “separate” to “the separate” and “service” to “Services”.

In lines 225-227: We corrected in the sentences the following: “included” to “shown”, and “This way of representation” to “This  representation”, and “generated data” to “data generated”, and “moments” to “the moment”, and “service” to “the service”, and “network” to “the network” and “denote” to “is denoted as”.

In lines 230-231: We corrected the title of Figure 5. to “Activation and repetition of the application services S1 to S4 and the communication interactions among network elements E1 to E8: (a) time-line; (b) time -plane”

(4.2) Multigraphs of data distribution in ITCN network traffic

Additional editing correction:

In lines 234-240: We corrected the sentences with “The data exchanged by the applicable service Srv at each moment of time t = (t0, t1, ...tN-1, tN, tm) are shown by presenting the network traffic as a multigraph (Figure 6a). The single-service multigraph (labeled SSMG_Srv_Adt) shows the amount of data Adt (kbps or Mbps) exchanged between  the network elements Ei and Ej (i ≠ j) by the application service Srv at time t. A single edge between the nodes (simple graphs) Ei and Ej (i ≠ j) represents the communication interaction between these network elements, where the amount of data Adt are distributed through the application service Srv at time t.”

In lines 243-249: We corrected the sentences to “The total data exchanged between the nodes Ei and Ej (i ≠ j) through all application services Srv = (S1, S2...Sn) at time t represents the unification of all the single-service multigraphs formed previously into one multi-service multigraph (labeled MSMG_S1Sn_Adt), as shown in Figure 6(b). The multi- service multigraph enables the definition of network traffic among the ITCN’s network elements at the observed moments of time t = (t0, t1, ...tN-1, tN, tm) and enables the application of graph sampling theory to perform predictions, as in [22].”.

In lines 250-251: We replaced the wholeFigure 6. in one row.

In lines 253-254: We corrected the title of Figure 6. “Data exchange multigraph among network elements E1 to E8 with the application services S1 to S2 at time t0 (a) single-service multigraph; (b) multi-services multigraph.”

In lines 255-259: We corrected in the sentence the following: “creating” to “creation”, and “presentation” to “the presentation”, and “communication” to “the communication”, and “Multi-services” to “multi-service” and “between” to “among the”.

(4.3) Matrix associated with the ITCN network traffic distribution multigraph

Additional editing correction:

In line 260: Editing correction in the title of subsection “with ITCN” to “with the ITCN”.

In lines 267-271: We corrected the sentences to “where Adijt is the amount of data distributed in the communication interactions between the nodes Ei and Ej (i≠j) with the application service Srv= (S1, S2, Sn) at the moment of time t = (t0, t1, ...tN-1, tN, tm). Figure 7 shows the single-service multigraph for data exchanged among the network elements E1 to E8 with the application service S1 at the moment of time t0, and its associated symmetric 8 x 8 matrix.”.

In line 273: We corrected the title of Figure 7. to “Data exchange single-service multigraph and its symmetric 8 x 8 matrix.”

In lines 274-275: In the sentence we replaced “time point” to “moments of time”.

Additional editing correction:

In line 283: We corrected in the sentence “of the value” to “in the value” and “distributed data between network elements ITCN” to “data distributed among the network elements of the ITCN.”.

In lines 288-291: To clarify the meaning we redefined the sentences to “For the multi-service multigraph, the associated symmetric n x n matrix TMSMG_S1Sn_Adt of data distribution shown in (7) is formed. The value of the distribution function of the total amount of data distributed through all application services Srv at the moments of time t = (t0, t1, ...tN-1, tN, tm) is defined by the associated symmetric matrix TMSMG_S1Sn_Adt.”

In line 294: We corrected in the sentence “all activated” to “all the activated” and “in moments” to “at the moments”.

In line 295: At the end of the sentence we added “where”.

In line 301: We corrected in the sentence “data amount distribution” to “amount of data distributed and “all applicaton” to “all the application”.

In line 302: At the end of the sentence we added “such that”.

In lines 305-307: We corrected in the sentence “all sets of associated multiple service” to “all the sets of the associated multi-service”.

(5) Distribution function of data distribution in ITCN by sampling multigraphs (replaced with “Generating the data distribution function in the ITCN by sampling multigraphs”)

Additional editing correction:

In lines 310-312: We corrected in the sentences the following: “the formed single-service” to “the single-service”, and “multigraphs” to “the multigraphs”, and “sampling associated” to “sampling the associated”.

In lines 320-325: We corrected in the sentence “Matrix” to “The matrix” and “at moments” to “at the moments” and “between network” to “between the network” and “distribution function” to “The distribution function”.

In lines 332-336: To clarify the meaning we redefined the sentence to “The sum of row margin (di) of the matrix (n x n), denoted d(2), d(3) .. d(i), and the updated row margins of the (n-1) x (n-1) submatrix are determined for the matrix TEi(t). The procedure of sampling and removing the matrix columns in TSSMG(t) is repeated until all the columns (c1, c2...cn) have been sampled. The value of each margin of the row (di) and the total margin of the matrix (M) is calculated according to the following:”.

In line 350: We corrected in the sentence “obtainned expression” to “expression obtained”.

In line 356: We corrected in the sentence “Combining expressions” to “Combining the expressions in”.

In line 363: We corrected in the sentence “cv2 calculates” to “cv2 is used to calculate”.

In lines 380-381: We corrected in the sentence “the formed set of values {q(TSSMG(t))} enables” to “the set of values {q(TSSMG(t))} thus formed enables”.

In lines 384-386: To clarify the meaning we redefined the sentence to “The graph of the data distribution function pdF (Figure 8) shows the regularity of the time of the change in the amount of data for distribution among the ITCN’s network elements through the application service Srv in the communication time interval ΔT= [t0, tm].

In line 390: We corrected in the sentence “procedure defines” to “procedure is used to define”.

In line 392: We corrected in the sentence the following: “the function pdF(Srv(t)) with the graphs” to “the function pdF(Srv(t)) to the graphs” and “allows the identification of” to “allows one to identify” .

In lines 395-397: We corrected in the sentence the following: “Identification” to “The identification”, and “of parameter” to “of the parameter” and “from graphs, figure 8 ” to “from the graphs (Figure 8).”

In line 398: We corrected in the sentence “OPNET software allows the import of graphics” to “OPNET software allows one to import graphics”.

In lines 399-401: To clarify the meaning we redefined the sentence to “This way, the distribution function can be used as a newly defined distribution for realization of the ITCN simulation model.”

(6) Conclusion and further research

Additional correction by the authors:

In lines 403-405: To clarify the meaning we redefined the sentences to “The application of multigraphs for describing data distribution in an ITCN is described in this study. It enables a more accurate definition of communication events in the network and a mathematical description of the network traffic. “.

In lines 410-412: To clarify the meaning we redefined the sentence to “Achieving more accurate results of simulating the predicted communication in an ITCN enables the integration of all the derived distribution functions for a description of the network traffic into the OPNET simulation model.”.

In line 412: We corrected in the sentence the following: “Based on” to “On the basis of” and “simulations ITCN” to “simulations of an ITCN” and “of distribution function” to “of the distribution function” and “data distribution” to “of data distribution”.

In lines 415-417: To clarify the meaning we redefined the sentence to “For realizing the simulation of network traffic flows, we propose the use of multi-service multigraphs, and the creation and use of data distribution functions pdF(S1Sn(ΔT)) of all services at the same time.”.

In lines 417-418: We corrected in the sentence “the continuation of the research we analyze the correlacion of data” to “future research, we will analyze the correlation of the data“.

In lines 420-421: We corrected in the sentence the following: “enable the prediction” to “enable predictions” and “of ITCN” to “of an ITCN”.

Reviewer 2 Report

This paper aims to present a new method for defining the data distribution function in a special purpose communication network, using multigraphs. The proposed method may be valuable to researchers. However, the lengthy sentences and countless grammatical errors make the comprehension of the actual method very difficult. The authors are highly encouraged to seek professional editing services, as these errors might take away the attention of the reader from the actual content entirely.

Some suggestions for the authors are as follows:

1.     Please declare the main section containing your method in its heading, so that the proposed method is easier to find.

2.     Line 85-88: This segment is ambiguous. I believe the authors intended to say that earlier methods consisted of complex procedures for describing and defining the network types. However, these sentences are worded in a way that the actual meaning is difficult to grasp. Please rephrase this segment.

3.     Line 142: “Amount of data for network distribution in time” does this heading mean the data distribution in the network over time? Please rephrase this heading to be more concise.

4.     Lines 151-152: This sentence is unclear. Does increasing the amount of data increase the amount of digital code required for protection? Please rephrase this sentence so that it is easier to understand.

5.     Lines 182-185: This sentence is unclear. Does simulating the incorrectly predicted network traffic in ITCN cause an inappropriate allocation function? Please rewrite this sentence.

6.     Lines 194-195: Unclear sentence. Please rephrase.

7.     Line 200-201: This sentence is ambiguous. Does changing the time interval values of the communication enable the mentioned definition?

8.     Line 207: the second instance of  seems to be redundant.

9.     Line 222: “Ad” -> Adt”

10. Line 262-265: The order of the words and the phrasing of this sentence make it extremely confusing. Please rephrase the sentence.

11. It appears that the word “distribution” has been used in two contexts: “data distribution” and “the transmission of data”. Using this word for both contexts will confuse the reader. Please use an alternative for the use of distribution in the “transmission” context.

12. Lines 408-410: “The method of applying multigraphs is also possible in networks with stochastically generated network traffic, while defining the variation in the amount of data to be generated.” Please rephrase this sentence to be more concise.

13. Lines 107-108: This sentence is unclear. Is the method executed on the mentioned example?

14. Please move “Table 1” so that it is positioned after the text where it is referenced.

15. Figure 1 falls out of the page. Please shift the figure more to the left so that it can be seen entirely.

16. Line 230: The figures and their legends do not align correctly. Also, the fonts used in the figures are too small.

Please be mindful of the grammatical errors and word choices:

17. Some words are repeated too many times (one example being “enable”, “allow”, and “realize”). Please use some alternatives.

18. Line 42: “communication network is a credible description of” -> “generating a credible description”

19. Line 89: “technological” -> “technologically”

20. Line 95: “defined or using” -> “defined by using”

21. Line 97: “of the type of network traffic” -> “network traffic type”

22. Line 101: “software” -> “softwares”

23. Lines 106-107: “the use of multigraphs describes” -> “we use multigraphs to describe”

24. Line 109: “a significant role for network design” -> “a significant role in network design”

25. Line 112: “to perform a mathematical model from the network”: I believe that “obtain” may be a better word choice.

26. Line 127: “time moments” -> “moments of time”

27. Line 143: “designing of a” -> Remove “of”.

28. Line 144: “that is sent” -> Remove “that is”.

29. Lines 150-151: “the encryption of” -> “encrypting”.

30. Line 156: “data to be transmitted” -> Remove “to be”.

31. Line 160: “to send network elements ITCN” -> “to send to the network elements in ITCN”

32. Line 161: Remove “time”.

33. Lines 170-171:  “the variation in the change in the amount of data over time” remove “in the change”

34. Lines 171-172: “defining the values of its parameters” can be simplified: “defining its parameters”.

35. Line 172: “The selection of the distribution …” This sentence is very similar to the one in the line above. I suggest adding “also” to the second sentence.

36. Line 179: “The time when the amount of data is generated” -> The data generation time.

37. Line 180: “That time of data generation …”> “The time of data generation”.

38. Line 189: “is derived” seems to be redundant.

39. Lines 191-192: I suggest moving “the network data distribution in ITCN is defined” to the beginning of the sentence.

40. Line 198: “joins to”-> is joined with

41. Line 199: “calculates by” -> “is calculated by”

42. Line 202: remove “of”.

43. Lines 240-241 and 255: “The creating” -> “The creation”

44. Line 262: “the associated” seems redundant.

45. Line 283: “ minimum and maximum-based value” -> “minimum and maximum values”

46. Line 308: Distribution function of data distribution” -> “Generating the data distribution function”

47. Line 310: “simulation OPNET model” -> “OPNET simulation model”

48. Line 318: “ is defined by the use of graphs and weighting coefficients by the method given in” too many “by”s.

49. Line 325: “The estimate of the value” -> “The estimated value”

50. Line 343: Insert comma between “For forming submatrices” and “number”.

51. Lines 359-360: “determines the evaluation of the” -> “evaluates the”

52. Line 497: “Review of comparison of characteristics of methods” can be simplified to “Review of the network traffic-defining methods”.

The formulations need to be consistent throughout the paper. In many cases, subscripts have not been applied correctly and there are many inconsistencies. Some examples are:

53. Line 149: It appears that the subscript is not rendered correctly. This also applies to everywhere  is referenced. Other instances of  have subscripts.

54. Line 267: “Adijt”: please use proper formatting and use subscripts.

55. Line 335: Please use correct formatting for “di” and use subscripts.

Throughout the paper, the sentences are very long and incomprehensible. Some examples of such long sentences are:

56. Lines 30-33.

57. Lines 47-49: I would also suggest changing “is the definition of the mathematical model” to “is defining the mathematical model”.

58. Lines 101-103: This sentence can be worded more clearly. I think the authors meant to say that using the other, older methods may lead to incorrect selection or description.

59. Lines 162-165: The sentence is too long and uses too many “of”s.

60. Lines 278-280.

61. Lines 368-371.

62. Lines 371-374.

Author Response

The authors are gratefull for the comments of the reviewers that enormously helped quality improvement of the work and this notification will be added in the acknowledgement. As requested, comments of reviewers are provided in received order and carefully answered to the best of authors knowledge with clear notification (remarked as yellow) what has been included in the edited manuscript:

Reviewer II Comments:

The following issues need to be looked at for further improvement of the paper.

Article

Additional editing correction by the authors:

In line 2: The title of the paper editing correction “Multigraphs” to “A multigraph”

Abstract

Additional correction by the authors:

In lines 11-16: To clarify the meanings, we reformulated the sentences to” We present a method based on multigraphs to mathematically define a distribution function in time for the generation of data exchange in a special purpose communication network. This is needed in the modeling and design of a communication network (CN) consisting of integrated telecommunications and computer networks (ITCN). Simulation models require precise definition of the network traffic communication. “

In line 16: Editing correction “the description of” to “describing the”.

In lines 19-20: To clarify the meanings, we reformulated the sentences to” The application of multigraphs enables the time and quantity of the data distribution to be displayed on the basis of operational procedures for the special- purpose communication unit.”

In lines 20-23: We corrected in the sentence “The multigraph” to “A multigraph” and “to be joined” to “to be defined”.

(1) Introduction

Answer: For the introduction improvement we conducted following changes:

Correction according Reviewer II Comment 56.

In lines 30-33: Two shorter and clearer have replaced the introductory sentence „The design of communication networks as a spatially distributed integrated telecommunication and computer network (ITCN) has been improved by application of computer simulation. Defining a simulation model of an ITCN realized by using advanced simulation software with integrated tools. These tools allow an analysis of the network elements’ parameters. “

Additional correction by the authors:

In lines 34-36: To clarify the meanings, we reformulated the sentences to” The application of the methodology of the ITCN simulation model design requires a precise definition of the network traffic in addition to a definition of the active and passive elements of the architecture and network topology [1].”

Correction according Reviewer II Comment 18.

In lines 41-43: To clarify the meanings, we corrected the sentence to “The problem in designing a simulation model is generating an accurate description of this network traffic.”.

Correction according Reviewer II Comment 57.

In lines 47-49: Instead of “is the definition of the mathematical model” we replaced to “is defining the mathematical model”. To describe an additional problem in the description of network traffic, we replaced it “certain types and quantities of data are realized” with “amount of data is realized”. After correction the additional problem of network traffic description is listed with “An additional problem of describing network traffic in the simulation model is defining the mathematical model according to which the generation of and change in the amount of data is realized.”

Additional correction by the authors:

In lines 49-50: To clarify the meaning, we removed from the sentence “the definition of”.

In line 59: We replaced the term “paper” with “study”.

In lines 57-58: To clarify the meaning in subsection, we removed the sentence “It is indicated the recognition of the basis for defining the data distribution in the network matrix implemented in the communication network”.

In lines 60-65: To clarify the meaning of the contribution we divide the first sentence on two, and we reformulated with “A new method of defining network traffic is proposed. The distribution function for creating a simulation model of a communication network is developed, based on the description of communication events and the values of the parameters they determine. The application of the method enables us to solve the problem of describing the time of data generation and distribution in the communication networks”. From the sentence in lines 64-65 we removed “and networks of similar purpose”.

In lines 66-69: To simplify the meaning of the contribution in the sentence we corrected “in comparison” to “and compared, and we removed “measurements of” and “in similar networks”. The long sentence we reformulated with “The application of multigraphs for the mathematical derivation of a more precise distribution function of data is proposed and compared with other methods in which the distribution function of data is approximated by the type of network traffic and by the time variation of the data.”.

In lines 70-75: We reformulated the sentence with “The application of multigraphs and their related matrices enables multiple descriptions of network traffic in terms of events and communication parameters, which enables their change in time to be mathematically represented as a function of the schedule. The new approach enables a more accurate description of the network traffic in the design of a simulation model of the communication network and time-accurate results in the simulation.”

Correction according Reviewer II Comment 1.

In lines 76-83: We have declared Section IV as the main one, which contains the concept and essence of the method.

We additionally corrected some terms and that segment to “The study is organized as follows. Section II provides an overview of the different methods used for defining and statistically describing network traffic. Definitions of all the starting elements needed to describe network communications are presented in Section III.  In Section IV, the basic concept and details of the proposed method of applying multigraphs for describing the time of network traffic and data distribution are presented. Section V presents an application of the mathematical derivation and a graphical representation of the time distribution functions in the proposed method. Section VI concludes the study and gives directions for further research in the application of the method.”.

(2) Related work

Answer: To clarify the meanings, we redefined the sentences and corrected the terms with more precise ones.

Correction according Reviewer II Comment 2.

In lines 85-88: This segment we rephrased to “In earlier works, different methods of defining network traffic were proposed. Network traffic is a complex time stochastic or deterministic process of network structure. Earlier methods consisted of complex procedures for describing and defining the network traffic. The methods are complex, especially for describing network traffic with the distribution of multiple data formats.”

Correction according Reviewer II Comment 19.

In line 89: The term we removed “technological” with the previous change.

Correction according Reviewer II Comment 20.

In line 95: To more clearly reference the use of the self-similarity method we corrected in the sentence with “statistical typification of network traffic with known distribution functions with parameter variation was defined or the traffic was described by using self-similarity related to heavy-tail distributions” and we added the new reference “[23]”.

Correction according Reviewer II Comment 21.

In line 97: We corrected “of the type of network traffic” to “network traffic type”. To clarify the meanings, we corrected the sentence with “Further definitions of network traffic were limited to the recognition network traffic type (voice, internet, HTTP, VoIP, multimedia, etc.) and descriptions of the intended distribution function.”.

Additional correction by the authors:

In lines 98-99: We correctef in the sentence “papers” to “studies” and “traffic is given” to “traffic are given”.

Correction according Reviewer II Comment 14.

From line 479: We have moved Table 1 after the text at the end of the subsection where it is referenced.

Correction according Reviewer II Comment 22.

In line 101: We corrected “software” to “software simulation packages”.

Correction according Reviewer II Comment 58.

In Lines 101-103: To describe the meanings, we redefined the sentences and replaced this segment with

“The application of the methods described in previous research may lead to incorrect selection or incorrect description of the statistical parameters of the distribution function. The consequence is that one may obtain incorrect simulation results and derive erroneous conclusions and decisions about the design of the network structure.”

Correction according Reviewer II Comment 23.

In lines 106-107: We corrected, “the use of multigraphs describes” to “we used multigraphs to describe”.

Correction according Reviewer II Comment 13.

In lines 107-108: To clarify the meaning we redefined the sentence to “This method was executed and tested on the example of deterministic arranged communication in the network”.

Correction according Reviewer II Comment 24.

In line 109: We corrected “a significant role for network design, network traffic design” to “a significant role in network design and we corrected “analysis of results” to “analysis of the results,”.

In lines 109-111: To clarify the meaning we reformulated the sentence to “The network traffic matrix model has a significant role in network design, network traffic design and analysis of the results, as in the method given in [17], [18].”

Correction according Reviewer II Comment 25.

In line 112: We corrected, “perform” to “obtain”.

Additional correction by the authors:

In lines 113-116: To clarify the meaning we corrected the sentence to “By applying the mathematical models given in [19],[20], the time dimension of the multigraph was added and the estimated distribution function that describes the network traffic as a statistical time event can be obtained.”

(3) Data exchange in the communication network

Answer: To clearly describe the meanings, we redefined the sentences and replaced the terms with more precise ones.

Additional correction by the authors:

In lines 118-119: To clarify the meaning we reformulated the sentence to “Central  to definition of network traffic models is the matrix of network traffic between the source and the destination of communications in the network.”

In lines 121-122: We corrected in the sentence “Realization” to “The realization” and “destination” to “the destination”.

Correction according Reviewer II Comment 26.

In line 127: We corrected in the sentence “time moments” to “moments of time”. This correction was made in the rest of the text of the paper. To clarify the meaning we additionally corrected “exchange” to “ exchanged” and “certain” to “the particular”.

Additional correction by the authors:

In lines 132-133: We corrected in the sentence “relations” to “relationships” and “ traffic is obtaind” to “ traffic can be obtained”.

In line 134: We corrected in the sentence “concept” to “concepts” and “exchange is defined” to “exchanges are defined”.

Correction according Reviewer II Comment 15.

In line 139: Figure 1 is centered.

Additional correction by the authors:

In line 140: Editing correction “of mapping” to “of the mapping” and “of network” to “of the network”.

(3.1) Amount of data for network distribution in time (replaced with “The data of network distribution over time”)

Correction according Reviewer II Comment 3.

In line 142: In order to bring the paragraph into relation with data exchange, we have redefined the title of the subsection with “The data of network distribution over time”.

Correction according Reviewer II Comment 27. and 28.

In lines 143-144: In the sentence we corrected “designing of a simulation” to “ designing a simulation ” and we removed “that is”. We corrected the sentence to “Designing a simulation model requires use to define the variation in the amount of data generated at the source sent to the destination. ”

Additional correction by the authors:

In lines 144-146:In the sentence we corrected “for exchange between elements” to “exchanged between the elements ”.

In line 147-151: To clarify the meaning we redefined the sentences to “The distribution of the amount of data occurs at the moments of time t = (t0, t1, ...tN-1, tN, tm).”.

Correction according Reviewer II Comment 53.

In line 149: No subscript is planned in the Ei and Srv numbering.

Correction according Reviewer II Comment 29.

In lines 150-151: In the sentence, we corrected “the encryption of” to “encrypting”. To clarify the meaning we redefined the sentence to “The transfer of information to the ITCN requires encrypting of the communication channels.

Correction according Reviewer II Comment 4.

In lines 151-152: To clarify the meaning we reformulated the sentence to “The amount of data for distribution in the network can be increased by the amount of digital code required for protecting information (reconstruction, encryption, error detection).”

Correction according Reviewer II Comment 30.

In line 156: In the sentence, we corrected “data to be transmitted” to “data to be transferred” and we removed “between source and destination”. After correction, the sentence is “The choice of the access technique, the technology and transmission medium, the communication protocols, and the data packet size (MTU) affect the amount of data to be transferred by the telecommunication links in the ITCN.”

Additional correction by the authors:

In line 157: To clarify the context meaning in the sentence we removed” to be transmitted” and we redefined the sentence with “The total data payload for distribution by the network from source to destination is determined by the steps in the procedure shown in Figure 2.”.

Correction according Reviewer II Comment 31.

In line 160: In the caption of Figure 2. we corrected “to send network elements ITCN” to “to send to the network elements in the ITCN”.

Correction according Reviewer II Comment 32.

In line 161: We removed “time” and the whole sentence has been moved to a new position before Figure 1.

Correction according Reviewer II Comment 59.

In lines 162-165: To clarify the meaning we redefined the sentence to “The distribution function of communication interactions between network elements according to the method given in [2] represents the law of data generation over time. Data generation is realized by the application services.”

(3.2) Distribution function for data amount variations (Replaced with “Distribution function for variations in the amount of data”)

Additional correction by the authors:

In line 167: We have corrected the title of the subsection with “Distribution function for variations in the amount of data”.

Correction according Reviewer II Comment 33.

In lines 170-171: We removed “in the change”.

Correction according Reviewer II Comment 34.

In lines 171-172: In the sentence, we corrected “defining the values of its parameters” to “one to define its parameters”.

Correction according Reviewer II Comment 35.

In line 172: We added in the second sentence “Moreover,” and we corrected “recording” to “record”.

Additional correction by the authors:

In line 178: We added the new reference “[23]”

Correction according Reviewer II Comment 36.

In line 179: We corrected in the sentence “The time when the amount of data is generated” to “The data generation time”.

Correction according Reviewer II Comment 37.

In line 180: In the sentence, we corrected “That time” to “The time” and “adjusted with” to “adjusted by”.

Correction according Reviewer II Comment 5.

In lines 182-185: To clarify the meaning we corrected the sentence to “If one chooses an inappropriate distribution function, or by incorrectly defining the value of the variation in the amount of data, or by incorrectly defining the time, the network traffic will be described incorrectly. Simulations of incorrectly described network traffic will not match the predicted network traffic in an ITCN.”

In lines 185-186: In accordance with the given comment we corrected the sentence to “In that case, the simulation results are not accurate for analyzing and optimizing the communication network.”.

Correction according Reviewer II Comment 38.

In line 189: We removed from the sentence “is derived”.

(4.) Description of the ITCN network distribution using multigraphs

Additional editing correction:

In line 190: Editing correction in the title of subsection “of ITCN” to “of the ITCN”.

Correction according Reviewer II Comment 39.

In lines 191-193: After correction, the sentence is “The network data distribution in an ITCN is realized on the basis of the operating procedures and according to the methodology specified in [1]. The information flows of the distribution of the predicted types of information between the network elements are described as well.”

Correction according Reviewer II Comment 6.

In lines 194-195: To clarify the meaning we redefined the sentence to “The multigraphs are defined on the basis of this description.”.

Additional correction by the authors:

In line 195: We corrected in the sentence “relations” to “relationships” and “of operational” to “of the operational”.

In line 197: We corrected in the sentence “data is exchanged” to “data are exchanged,”

Correction according Reviewer II Comment 40.

In line 198: In first sentence, we corrected “joins to” to “is joined with”.  

Additional correction by the authors:

In line 198: To clarify the meaning we redefined the second sentence to “The corresponding value of the distribution function for each moment of time is calculated by mathematical estimation of the similarity matrix associated with the multigraph.”

Correction according Reviewer II Comment 41.

In line 199: In first sentence, we corrected “calculates by” to “is calculated by”. 

Correction according Reviewer II Comment 7.

In line 200-201: To clarify the meaning we redefined the second sentence to “The use of all the calculated values for all moments of time in the communication interval ΔT = [t0, tm] enables the definition of the appropriate distribution function.”

(4.1) Data distribution time scheme between ITCN network elements

Correction according Reviewer II Comment 42.

In line 202: In the title of the subsection, we removed “of”.

Additional editing correction:

In lines 203-204: We corrected in the sentenece “Generation” to “The generation” and “ITCN” to “the ITCN” and “network” to “the network”.

Correction according Reviewer II Comment 8.

In line 207: We removed from the sentence “Ej”.

Additional editing correction:

In line 209: We corrected in the sentenece “the transformation” to “ transformation”.

In line 211: We corrected in the sentenece “operational ” to “the operational”.

In line 213: We corrected in the title of Figure 3. “Time-Line” to “Time-line” and “network” to “the network”.

In lines 215-216: We corrected in the sentence “time schemes” to “Time-line” and “time of activation t = (t0, t1...tN-1, tN,tm)” to “ moments of time t = (t0, t1...tN-1, tN,tm) of activation”.

In lines 216-217: We removed “which is”.

In lines 218-219: We corrected in the sentence “separate” to “the separate” and “service” to “Services”.

Correction according Reviewer II Comment 9.

In line 222: We corrected “Ad” to “Adt”.

Additional editing correction:

In lines 225-227: We corrected in the sentences the following: “included” to “shown”, and “This way of representation” to “This  representation”, and “generated data” to “data generated”, and “moments” to “the moment”, and “service” to “the service”, and “network” to “the network” and “denote” to “is denoted as”.

Correction according Reviewer II Comment 16.

In line 230: Figure 5. is centered and the font is enlarged as much as possible.

Additional editing correction:

In lines 230-231: We corrected the title of Figure 5. to “Activation and repetition of the application services S1 to S4 and the communication interactions among network elements E1 to E8: (a) time-line; (b) time -plane”

(4.2) Multigraphs of data distribution in ITCN network traffic

Additional editing correction:

In lines 234-240: We corrected the sentences with “The data exchanged by the applicable service Srv at each moment of time t = (t0, t1, ...tN-1, tN, tm) are shown by presenting the network traffic as a multigraph (Figure 6a). The single-service multigraph (labeled SSMG_Srv_Adt) shows the amount of data Adt (kbps or Mbps) exchanged between  the network elements Ei and Ej (i ≠ j) by the application service Srv at time t. A single edge between the nodes (simple graphs) Ei and Ej (i ≠ j) represents the communication interaction between these network elements, where the amount of data Adt are distributed through the application service Srv at time t.”

Correction according Reviewer II Comment 43.

In lines 240-241 and 255: We corrected the sentence to “The creation of all the single-service multigraphs between the network elements Ei and Ej (i≠j) through the application service Srv for each moment of time t = (t0, t1, ...tN-1, tN, tm) enables the presentation of data exchanged during the communication time interval ΔT = [t0, tm].”.

Additional editing correction:

In lines 243-249: We corrected the sentences to “The total data exchanged between the nodes Ei and Ej (i ≠ j) through all application services Srv = (S1, S2...Sn) at time t represents the unification of all the single-service multigraphs formed previously into one multi-service multigraph (labeled MSMG_S1Sn_Adt), as shown in Figure 6(b). The multi- service multigraph enables the definition of network traffic among the ITCN’s network elements at the observed moments of time t = (t0, t1, ...tN-1, tN, tm) and enables the application of graph sampling theory to perform predictions, as in [22].”.

In lines 250-251: We replaced the wholeFigure 6. in one row/

In lines 253-254: We corrected the title of Figure 6. “Data exchange multigraph among network elements E1 to E8 with the application services S1 to S2 at time t0 (a) single-service multigraph; (b) multi-services multigraph.”

In lines 255-259: We corrected in the sentence the following: “creating” to “creation”, and “presentation” to “the presentation”, and “communication” to “the communication”, and “Multi-services” to “multi-service” and “between” to “among the”.

(4.3) Matrix associated with the ITCN network traffic distribution multigraph

Additional editing correction:

In line 260: Editing correction in the title of subsection “with ITCN” to “with the ITCN”.

Correction according Reviewer II Comment 44.

In lines 262-265: We removed “associated” and to clarify the meaning we redefined sentences to “Mathematically, the multigraph data distribution in network traffic of the ITCN is represented by the symmetric matrix TSSMG_Srv_Adt in (1) with integer terms and a diagonal of zero, where n is the number of network elements Ei. The associated symmetric matrix is formed by using a time-line or time-plane of the communication interactions (Figure 5), or by using a single-service multigraph (Figure 6a), such that”

Correction according Reviewer II Comment 54.

In line 267: We formatted “Adijt” to “Adijt”.

Additional editing correction:

In lines 267-271: We corrected the sentences to “where Adijt is the amount of data distributed in the communication interactions between the nodes Ei and Ej (i≠j) with the application service Srv= (S1, S2, Sn) at the moment of time t = (t0, t1, ...tN-1, tN, tm). Figure 7 shows the single-service multigraph for data exchanged among the network elements E1 to E8 with the application service S1 at the moment of time t0, and its associated symmetric 8 x 8 matrix.”.

In line 273: We corrected the title of Figure 7. to “Data exchange single-service multigraph and its symmetric 8 x 8 matrix.”

In lines 274-275: In the sentence we replaced “time point” to “moments of time”.

Correction according Reviewer II Comment 60.

In lines 278-280: To clarify the meaning we redefined the sentence to “The set of associated matrices TSSMG_Srv_Adt enables one to define the function for the distribution of data in the network through the service Srv = (S1, S2...Sn) in the communication time interval ΔT = [t0, tm].”.

Correction according Reviewer II Comment 45.

In line 283: We corrected in the sentence “minimum and maximum-based value” to “minimum and maximum values”.

Additional editing correction:

In line 283: We corrected in the sentence “of the value” to “in the value” and “distributed data between network elements ITCN” to “data distributed among the network elements of the ITCN.”.

In lines 288-291: To clarify the meaning we redefined the sentences to “For the multi-service multigraph, the associated symmetric n x n matrix TMSMG_S1Sn_Adt of data distribution shown in (7) is formed. The value of the distribution function of the total amount of data distributed through all application services Srv at the moments of time t = (t0, t1, ...tN-1, tN, tm) is defined by the associated symmetric matrix TMSMG_S1Sn_Adt.”

In line 294: We corrected in the sentence “all activated” to “all the activated” and “in moments” to “at the moments”.

In line 295: At the end of the sentence we added “where”.

In line 301: We corrected in the sentence “data amount distribution” to “amount of data distributed and “all applicaton” to “all the application”.

In line 302: At the end of the sentence we added “such that”.

In lines 305-307: We corrected in the sentence “all sets of associated multiple service” to “all the sets of the associated multi-service”.

(5) Distribution function of data distribution in ITCN by sampling multigraphs (replaced with “Generating the data distribution function in the ITCN by sampling multigraphs”)

Correction according Reviewer II Comment 46.

In line 308: We have redefined the title of the subsection with “Generating the data distribution function in the ITCN by sampling multigraphs”

Correction according Reviewer II Comment 47.

In line 310: We corrected in the sentence “simulation OPNET model” to “OPNET simulation model”.

Additional editing correction:

In lines 310-312: We corrected in the sentences the following: “the formed single-service” to “the single-service”, and “multigraphs” to “the multigraphs”, and “sampling associated” to “sampling the associated”.

Correction according Reviewer II Comment 48.

In line 318: To clarify the meaning we redefined the sentence to “Additionally, the distribution function for the approximation of multigraphs is defined by using graphs and weighting coefficients and applying the methods given in [19] and [21].”

Additional editing correction:

In lines 320-325: We corrected in the sentence “Matrix” to “The matrix” and “at moments” to “at the moments” and “between network” to “between the network” and “distribution function” to “The distribution function”.

Correction according Reviewer II Comment 49.

In line 325: We corrected in the sentence “The estimate of the value” to “The estimated value”.

Additional editing correction:

In lines 332-336: To clarify the meaning we redefined the sentence to “The sum of row margin (di) of the matrix (n x n), denoted d(2), d(3) .. d(i), and the updated row margins of the (n-1) x (n-1) submatrix are determined for the matrix TEi(t). The procedure of sampling and removing the matrix columns in TSSMG(t) is repeated until all the columns (c1, c2...cn) have been sampled. The value of each margin of the row (di) and the total margin of the matrix (M) is calculated according to the following:”.

Correction according Reviewer II Comment 55.

In line 335: We corrected “di” to “di”. 

Correction according Reviewer II Comment 50.

In line 343: We corrected in the sentence “submatrices number” to “the submatrices, the number” and “and corresponds” to “which corresponds”.

Additional editing correction:

In line 350: We corrected in the sentence “obtainned expression” to “expression obtained”.

In line 356: We corrected in the sentence “Combining expressions” to “Combining the expressions in”.

Correction according Reviewer II Comment 51.

In lines 359-360: We corrected “determines the evaluation of the” to “evaluates the”.

Additional editing correction:

In line 363: We corrected in the sentence “cv2 calculates” to “cv2 is used to calculate”.

Correction according Reviewer II Comment 61.

In lines 368-371: To clarify the meaning we redefined the long sentence to “The application of the previous procedure to all single-service multigraphs and their associated matrices defines the change in the amount of data for distribution in the network. The change in the amount of distributed data is realized between network elements Ei and Ej (i ≠ j), through the application service Srv in the communication time interval ΔT = [t0, tm]. “.

Correction according Reviewer II Comment 62.

In lines 371-374: To clarify the meaning we redefined the long sentence to “The calculated values of the distribution function form a set of values of the distribution function q(TSSMG(t)). These values enable one to define the data distribution function pdF (Srv(t)) of the application service Srv in the communication time interval ΔT= [t0, tm]: “.

Additional editing correction:

In lines 380-381: We corrected in the sentence “the formed set of values {q(TSSMG(t))} enables” to “the set of values {q(TSSMG(t))} thus formed enables”.

In lines 384-386: To clarify the meaning we redefined the sentence to “The graph of the data distribution function pdF (Figure 8) shows the regularity of the time of the change in the amount of data for distribution among the ITCN’s network elements through the application service Srv in the communication time interval ΔT= [t0, tm].

In line 390: We corrected in the sentence “procedure defines” to “procedure is used to define”.

In line 392: We corrected in the sentence the following: “the function pdF(Srv(t)) with the graphs” to “the function pdF(Srv(t)) to the graphs” and “allows the identification of” to “allows one to identify” .

In lines 395-397: We corrected in the sentence the following: “Identification” to “The identification”, and “of parameter” to “of the parameter” and “from graphs, figure 8 ” to “from the graphs (Figure 8).”

In line 398: We corrected in the sentence “OPNET software allows the import of graphics” to “OPNET software allows one to import graphics”.

In lines 399-401: To clarify the meaning we redefined the sentence to “This way, the distribution function can be used as a newly defined distribution for realization of the ITCN simulation model.”

(6) Conclusion and further research

Additional correction by the authors:

In lines 403-405: To clarify the meaning we redefined the sentences to “The application of multigraphs for describing data distribution in an ITCN is described in this study. It enables a more accurate definition of communication events in the network and a mathematical description of the network traffic. “.

Correction according Reviewer II Comment 12.

In lines 408-410: To be more concise we redefined the sentence to “The method of applying multigraphs is also possible in networks with stochastically generated network traffic, which requires a definition of the variations in the amount of data to be generated between the source and destination. “.

Additional correction by the authors:

In lines 410-412: To clarify the meaning we redefined the sentence to “Achieving more accurate results of simulating the predicted communication in an ITCN enables the integration of all the derived distribution functions for a description of the network traffic into the OPNET simulation model.”.

In line 412: We corrected in the sentence the following: “Based on” to “On the basis of” and “simulations ITCN” to “simulations of an ITCN” and “of distribution function” to “of the distribution function” and “data distribution” to “of data distribution”.

In lines 415-417: To clarify the meaning we redefined the sentence to “For realizing the simulation of network traffic flows, we propose the use of multi-service multigraphs, and the creation and use of data distribution functions pdF(S1Sn(ΔT)) of all services at the same time.”.

In lines 417-418: We corrected in the sentence “the continuation of the research we analyze the correlacion of data” to “future research, we will analyze the correlation of the data“.

In lines 420-421: We corrected in the sentence the following: “enable the prediction” to “enable predictions” and “of ITCN” to “of an ITCN”.

Correction according Reviewer II Comment 52.

In line 479: We have corrected the title of the Table 1. “Review of comparison of characteristics of methods” to “Review of the network traffic defining methods”.

Round 2

Reviewer 1 Report

Authors have throughly considered all comments and suggestions,  and revised the paper accordingly 

Reviewer 2 Report

Thanks for considering the comments. The paper is now in good shape and can be published in the journal.